# PROTACs in Epigenetic Cancer Therapy: Current Status and Future Opportunities

**DOI:** 10.3390/molecules28031217

**Published:** 2023-01-26

**Authors:** Xuelian Liu, Anjin Wang, Yuying Shi, Mengyuan Dai, Miao Liu, Hong-Bing Cai

**Affiliations:** 1Department of Gynecological Oncology, Zhongnan Hospital of Wuhan University, Wuhan 430071, China; 2Hubei Key Laboratory of Tumor Biological Behaviors, Wuhan 430071, China; 3Hubei Cancer Clinical Study Center, Wuhan 430071, China; 4Department of Pathology, Brigham and Women’s Hospital, Harvard Medical School, Boston, MA 02115, USA

**Keywords:** epigenetics, cancer therapy, PROTACs, protein degradation

## Abstract

The epigenetic regulation of gene functions has been proven to be strongly associated with the development and progression of cancer. Reprogramming the cancer epigenome landscape is one of the most promising target therapies in both treatments and in reversing drug resistance. Proteolytic targeted chimeras (PROTACs) are an emerging therapeutic modality for selective degradation via the native ubiquitin-proteasome system. Rapid advances in PROTACs have facilitated the exploration of targeting epigenetic proteins, a lot of PROTAC degraders have already been designed in the field of epigenetic cancer therapy, and PROTACs targeting epigenetic proteins can better exploit target druggability and improve the mechanistic understanding of the epigenetic regulation of cancer. Thus, this review focuses on the progress made in the development of PROTAC degraders and PROTAC drugs targeting epigenetics in cancer and discusses challenges and future opportunities for the field.

## 1. Introduction

Epigenetics refers to heritable changes in gene expression and phenotype without altering the underlying deoxyribonucleic acid (DNA) sequence (genotype) [1]. In the past decades of research, epigenetic dysregulation has had a very close relationship with cancer progression [2,3,4]. The major epigenetic modifications include histone methylation, acetylation and phosphorylation, DNA methylation, and others, which take place in the intracellular compartment [5]. There are a variety of epigenetic target inhibitors entering clinical research, but most of them are limited to hematological tumors. It was hard to screen out candidate drugs with sufficiently safe potency for the conventional methods of drug development, such as monoclonal antibodies or small molecular inhibitors. Proteolytic targeted chimeras (PROTACs) is a kind of drug design that can be efficiently targeted for degradation by recruiting E3 ligases intracellularly without developing resistance and is also extremely promising in the field of cancer therapy [6,7]. In recent years, PROTAC drugs have been developed rapidly, and PROTACs targeting epigenetic genetics have emerged and performed extraordinarily in cancer research. In this article, we summarize the research of PROTACs in the field of epigenetic genetics in cancer.

## 2. Development of PROTACs

PROTACs are an emerging paradigm-shifting technology that uses the ubiquitin-proteasome system to achieve targeted protein degradation. As shown in Figure 1a, they consist of three parts: a ligand that targets the protein of interest (POI), a second ligand that binds the E3 ubiquitin ligase, and a linker that connects the two parts. PROTACs form a ternary complex between the E3 ligase and POI, and this induced proximity leads to the formation of polyB chains on the substrate and, ultimately, to POI degradation mediated by the proteasome [8]. Kruse and Deshays first proposed the groundbreaking idea of PROTACs in 2001. The resulting PROTAC-1 recruited methionine aminopeptidase-2 (MetAP-2) to the Skp1-Cullin-F box complex containing Hrt1 (SCF)β-TRCP for ubiquitination [9]. PROTAC1 represented the first proof of concept for this technology; however, such first-generation PROTACs used phosphopeptides as E3-binding components and were, therefore, unstable in vivo and not cell permeable, limiting their use as chemical probes [10]. In 2008, Ashley R. Smith developed a heterobifunctional all-small-molecule PROTAC with improved pharmacokinetic features capable of inducing proteasomal degradation of androgen receptors in tumor cells. This cell-permeable PROTAC consisted of a non-steroidal androgen receptor ligand (SAMM) and a mouse double minute 2 homolog ligand (Nutlin) linked by a polyethylene glycol (PEG)-based linker. Sarm-nutlin PROTAC recruited androgen receptors to the mouse double minute 2 homolog (MDM2), which acts as an E3 ubiquitin ligase. This resulted in the ubiquitination of the androgen receptor and its subsequent degradation by the proteasome [11]. Since then, a series of ligands for E3 ligases, including inhibitors of apoptotic proteins, brain proteins (CRBN), and von Hippel-Lindau (VHL), have been identified and used in the development of PROTAC [12,13,14] (Figure 1b). 

In 2013, the laboratory of J. T. Hines designed a PROTAC based on the VHL peptide ligand and demonstrated for the first time that a PROTAC preferentially inhibited ovarian cancer cell proliferation and tumor growth in a mouse model without significant adverse effects on animal health [15]. In 2019, Bavdegalutamide (ARV-110) became the first orally bioavailable small-molecule PROTAC drug for protein degradation targeting chimera to enter global clinical trials. Developed by Arvinas, ARV-110 selectively targeted androgen receptors for degradation. In February 2022, Arvinas presented its research results, which showed that ARV-110 had great potential in the treatment of prostate cancer. As of March 2022, 24 PROTACs had entered clinical trials.

## 3. Epigenetic Regulation in Cancer Therapy

Epigenetics was defined as the heritable changes in gene expression that would not be coded in the chromosome itself [1]. Epigenetic regulation is mediated by four main mechanisms: DNA methylation, histone post-translational modification, chromatin structure regulation, and non-coding ribonucleic acid regulation [16]. The development and progression of cancer in humans is thought to be driven by a combination of epigenetic and genetic alterations that result in the activation of a multistep cancer program [17]. The frequency of genetic alterations is generally low in most types of cancer, but epigenetic alterations exceed somatic mutations by far in almost all types of human cancers [18,19,20]. Moreover, epigenetic alterations have considerable plasticity and reversibility, which support their potential use as targets for cancer treatment [21]. In recent years, numerous studies have suggested that epigenetic alterations play critical roles in a wide range of cancer types. Cancer-associated epigenetic alterations in tumors reveal potentially reversible targets for existing drugs and for an increasing repertoire of new drugs [22,23,24,25]. At present, a variety of PROTAC drugs targeting the regulation of cancer epigenetic targets are being tested, potentially providing new directions and methods for cancer treatment.

## 4. PROTACs for Epigenetic Targets in Cancer

### 4.1. Histone Acetylation

Histones are the core components of the nucleosome subunit, which is surrounded by 147 base pairs of DNA fragments and forms an octamer containing four core histones (H3, H4, H2A and H2B). Histones are mostly globular and have a characteristic side chain or tail densely packed with basic lysine and arginine residues that are subject to extensive covalent post-translational modification.

Bromodomain-containing protein 4 (BRD4) is a crucial member of the bromodomain and extracellular domain (BET) family, which is characterized by two bromodomains at the N-terminal and one extracellular domain at the C-terminal. These two pressed can recognize and interact with acetylated lysine residues at the N-termini of histones; the extracellular domain has not yet been fully characterized but is generally considered to serve as a scaffold for recruiting a variety of transcriptional regulators [26]. BRD4 plays the role of the reader in epigenetics. BRD4 can recruit transcriptional regulators to specific genomic sites, thus playing a pivotal part in regulating gene expression [27,28,29]. Several PROTACs targeting BRD4 are currently available (Table 1).

#### 4.1.1. ARV-825

Many studies have shown that bromodomain and extracellular domain inhibitors (BETi) treatments involve reversible binding and incomplete inhibition of BRD4, which may impair BETi activity in cancer cells [77,78]. Lu et al. designed a heterobifunctional PROTAC, ARV-825; unlike BETi, the BET PROTAC ARV-825 recruits and utilizes an E3-ubiquitin ligase to efficiently degrade BRD4. This drug is the first to successfully block the E3 ligase CRBN using the clinically approved immunomodulatory drug pomalidomide. Cereblon is a widely expressed protein, so this approach may be applicable to various organs and/or tissues, as diverse target proteins recruit ligands for various diseases to attach [32]. In lymphoma and acute myeloid leukemia (AML), ARV-825 induces more apoptosis than BETi. ARV-825 induces high levels of apoptosis in ruxolitinib-persistent or ruxolitinib-resistant secondary acute myeloid leukemia (sAML) cells in vitro [33,34]. In cholangiocarcinoma cells, ARV-825 has been shown to significantly upregulate protein 21 expression in the G1 phase and prevent cell cycle progression, leading to the rapid and sustained degradation of BRD4 in cholangiocarcinoma [35]. In thyroid cancer, oral administration of ARV-825 effectively inhibited the growth of thyroid cancer cell TPC-1 xenograft tumors in severe combined immunodeficient mice, suggesting that ARV-825 inhibits thyroid cancer cell growth in vitro and in vivo [36]. In addition, in gastric cancer, multiple myeloma, neuroblastoma, triple-negative breast cancer and ovarian cancer, ARV-825 showed an unprecedented inhibitory effect on tumor cell growth [37,38,39,40,41,42]. In recent years, research on ARV-825 has led to some new in-depth findings. The fusion of ARV-825 and nanotechnology has enabled the targeting of multiple cancers to achieve better therapeutic effects and also provides a new treatment method for drug-resistant cells [43,44]. The massive infiltration of tumor-associated macrophages in gliomas hinders the efficacy of drugs; however, incorporating ARV-825 into PEG composed of substance P peptides therapeutic nanosystems constructed in composite micelles enables penetration of the blood–brain barrier to target brain tumors, attenuate cell proliferation, induce apoptosis, and inhibit M2 macrophage polarization, with consequent anti-tumor effects [45]. In antiretroviral drug-loaded PEG-polylactic-co-glycolic acid polymer nanoparticles in pancreatic cancer two-dimensional cell culture and three-dimensional multicellular tumor spheroids, the model can continue to explain that BRD4 shows a good anticancer effect and can effectively treat pancreatic cancer [46]. Experimental results have also shown excellent synergistic effects of dual ARV-825 and nintedanib-loaded PEGylated nanoliposomes in melanoma, providing a new approach for the treatment of vemurafenib-resistant melanoma [47]. Experiments with ARV-825-loaded nanoliposomes in lung cancer and hepatocellular carcinoma have shown that nano-fusion leads to better tumor penetration and lethality [48,49].

#### 4.1.2. MZ1

In 2015, Michael Zengerle’s team linked the broad BET-selective bromodomain inhibitor JQ1 and the ligand of the E3 ubiquitin ligase VHL to form the compound MZ1 for the reversible, durable, and selective removal of BRD4 [50]. In triple-negative breast and ovarian cancer cell lines, mez1 inhibits the growth of both sensitive and resistant cells and acts on allogeneic tumors in vitro [40]. MZ1 has exhibited excellent antitumor effects in B-cell-like diffuse large B-cell lymphoma, neuroblastoma, and colon cancer cells [51,52,53]. In Human epidermal growth factor receptor 2 (HER2) positive cells, the combined use of MZ1 and trastuzumab could enhance the induction of apoptosis, leading to better inhibition of tumors. On this basis, nanoparticles conjugated with trastuzumab were used to encapsulate mez1 and deliver it to HER2-positive breast cancer cells, resulting in the increased induction of cancer cells [54,55].

#### 4.1.3. ARV-771

Kanak Raina’s laboratory designed BET PROTAC ARV-771 based on the VHL E3 ligase. This was the first BET-targeted PROTAC to show efficacy against solid malignancies. In prostate cancer cells, ARV-771 could restrain androgen receptor signaling and androgen receptor levels to inhibit tumor growth [56,57]. Subsequently, in liver cancer, non-small-cell lung cancer, and esophageal squamous cell carcinoma cell lines, ARV-771 could inhibit the cell viability of tumor cells by preventing cell cycle progression and triggering apoptosis, resulting in the development of a cisplatin and radiological screening program for non-small-cell lung cancer, possibly creating the most promising anticancer drugs [58,59,60]. ARV-771 was also shown to form a folate protein complex with the folate group that could minimize potential toxicity in a tissue-selective manner [61].

#### 4.1.4. dBET1/6

BET1 is a PROTAC drug designed in 2016, which is a direct-acting inhibitor of BET bromodomain JQ1 and CRBN ligand binding and can inhibit the BET bromodomain. In human leukemia xenografts, dBET exerts antitumor effects by pharmacologically disrupting BRD4 production [62]. dBET1 shows accelerated apoptosis and anticancer effects against different types of acute myeloid leukemia cell lines [34]. Studies have shown that ponatinib can sensitize cells to dBET1 by enhancing apoptosis of cancer cells, resulting in a better targeted degradation effect [63]. Adding a light-controlled switch to dBET1 to induce limited degradation by ultraviolet A (UV-A) irradiation at a specific time and rate enables the use of PROTACs in precision medicine [64]. The discovery of dBET6, which was optimized for the structure of dBET1 and has high cellular permeability, also confirmed that BET degradation differs in mechanism and capacity from BET bromodomain inhibition [65]. dBET6 downregulates the expression of myelocytomatosis viral oncogene (Myc) in all cancer cell lines, has anti-tumor effects, and can also block cell resistance in most chronic myeloid leukemia lines, and is more effective than the first-generation BRD4-targeted drug dBET1 [66,67,68]. Studies have combined dBET6 and nanotechnology to create a novel multifunctional nanoprotein complex that could directly eliminate lung cancer cells, inhibit tumor growth, and reshape the tumor microenvironment [69].

#### 4.1.5. Other Histone Acetylation PROTACs

A1874 is a nutlin-based PROTAC designed by John Hiness’s team that targets BRD4 degradation. It could degrade 98% of BRD4 in cells with nanomolar efficiency and showed stable complementary ability with tumor protein 53 (p53). Cells were more effective than PROTACs using VHL [70]. A1874 has also been shown to effectively inhibit the proliferation, invasion, and migration of colon cancer cells. Oral administration of A1874 potently restrained colon cancer xenograft growth in severely combined immunodeficient mice [71].In addition, there are two BDD4 PROTAC QCA570and GNE-987 that are of great significance.QCA570, designed by Chong Qin’s team, is an extremely potent and efficient BET protein-targeted degrader that degrades the BET protein at low picomolar (pM) concentrations in leukemia cells and could achieve complete and durable regression of tumors in mice at a tolerated dose [72]. In non-small-cell lung cancer cells, a combination of QCA570 and osimertinib could effectively inhibit osimertinib-resistant EGFR-mutant cells, suggesting that QCA570 has potential applications in the treatment of acquired osimertinib resistance [73]. GNE-987, a ternary complex formed between BRD4B1,BRD4B2and VHL E3-ubiquitin ligase is a novel BRD4 degrader that has been shown to be more efficient than in vitro degradation products of PROTACs MZ1 and ARV-825 [74,75]. In addition to BRD4. The paragenic chromatin regulators CREB-binding protein (CBP) and protein 300 also play a very important role in histone acetylation. The CBP and p300 are vital factors in the establishment and activation of enhancer-mediated transcription [79]. In cancer, CBP/p300 is associated with oncogenes and tumor suppressors. dCBP-1 is a CBP-targeting PROTAC drug comprising a bromine-based structural inhibitor and CRBN E3 ligase ligand. In multiple myeloma, dCBP-1 could reduce the expression of Myc, significantly inhibit the proliferation of cells, and showed a superior anti-tumor effect [76].

### 4.2. HDACs

To date, 18 Histone deacetylase (HDAC) enzymes have been identified in mammalian cells and subdivided into four main categories based on their homology with yeast HDAC. Three of the four classes (classes I, II, and IV) are zinc-dependent enzymes, whereas Class III HDAC is a NAD-dependent histone deacetylase. Class I HDACs comprise HDAC1, HDAC2, HDAC3, and HDAC8. Class II HDACs comprise HDAC4, HDAC5, HDAC6, HDAC7, HDAC9, and HDAC10. HDAC11 is the only class IV HDAC representative. Class III HDACs, also known as Sirtuins (SIRT) 1–7, are homologous to the yeast Sirt protein Sir2. Sirtuins are widely expressed in human tissues and regulate various biological functions [80]. The dysregulation of HDAC enzymes has been demonstrated in many different cancer types [81]. Multiple HDAC-targeting PROTACs have been designed and shown to be effective against various cancer cell lines (Table 2).

#### 4.2.1. HDAC6

HDAC6 is a class IIb HDAC that is mainly expressed in the cytoplasm and is responsible for the deacetylation of cytoplasmic proteins and regulating the turnover of misfolded and polyubiquitinated proteins [91,92]. Overexpression of HDAC6 in multiple cancers is an indicator of advanced disease progression, higher rates of metastasis, and lower survival rates [93,94,95,96,97]. Yang Jia’s team designed the first PROTAC targeted degrader of HDAC6 by linking a pan-HDAC inhibitor to CRBN E3 ubiquitin ligand.However, this degrader was not selective for HDAC6. Tethering the selective HDAC6 inhibitor next-generation hydrofurotanone to a CRBN ligand resulted in a drug that could counteract the proliferation of multiple myeloma. Nevertheless, the versatility of CRBN-based HDAC6 degraders limited their use as specific probes for HDAC6-related cellular pathway studies [82,83,84]. Zixuan Ans team designed a series of HDAC6-selective PROTAC drugs. Among them, NP8 exhibited particularly good selective degradation of HDAC6 and inhibition of cell proliferation in multiple myeloma cells [85].

#### 4.2.2. HDAC3

HDAC3 belongs to the first class of HDAC enzymes, which are involved in G2/M progression in the cell cycle, gene replication, and gene damage response [98,99]. Several studies have shown that HDAC3 is associated with the progression of various cancers and may be a key factor in the progression of breast cancer, colorectal cancer, and pancreatic cancer metastasis. Its expression levels are negatively correlated with survival of cancer patients [100,101,102,103]. XZ9002, the first HDAC3-specific PROTAC, effectively reduces HDAC3 in breast cancer cells and is formed by the linkage of VHL E3 ligase and an inhibitor of HDAC3. Owing to their catalytic mechanism of action and isozyme selectivity, this new class of HDAC3-specific degraders can overcome the dose-limiting toxicity associated with traditional HDACi and are more efficient than traditional HDACi. They are thus more effective in inhibiting tumor growth. This may be critical for realizing the therapeutic potential of HDAC inhibition in the clinic [86]. Almost at the same time, another PROTAC based on the CRBN ligand and an anthranilide-based class I HDAC inhibitor was designed. It showed HDAC3 selectivity, with a better degradation effect on HDAC3 than on HDAC1 and HDAC2, and had a favorable effect in inhibiting inflammatory reactions [87].

#### 4.2.3. Sirtuin 2

Sirtuin 2 (SIRT2), a member of the third HDAC family, has a very significant role in the control of cell cycle progression and has been shown to be a checkpoint for mid/metaphase processes and G2/M transition. Studies have demonstrated that SIRT2 is involved in cancer metastasis and prognosis, which makes it a promising target for cancer therapy [104,105,106]. In 2018, a sirtuin rearrangement ligand (SirReals), as a highly potent and isoform-selective Sirt2 inhibitor conjugated to thalidomide, a bona fide cerebellar ligand, a PROTAC was developed to induce the degradation of Sirt2 in HeLa (Cervical cancer) cells and inhibit cell viability. On this basis, the team developed a PROTAC that used Parkin as an E3 ubiquitin ligase to target SIRT2 for degradation, providing a new direction for the development of PROTACs [88,89]. Subsequently, two PROTACs, TM-P2-Thal and TM-P4-Thal, were produced. They also used SirReals, and thalidomide, a ligand that recruits CRBN to synthesize targeted protein drugs. These two drugs can selectively degrade SIRT2 in breast cancer cells and effectively inhibit tumor proliferation [90].

### 4.3. Histone Methylation

Histone methylation is accomplished by histone methyl transferases. Methylation can occur on the lysine and arginine residues of histones; lysine residues can be mono-, di-, or tri-methylated, and arginine residues can be mono- or di-methylated. These varying degrees of methylation greatly increase the complexity of histone modification and regulation of gene expression [107]. Studies have shown that histone methylation occurs in pancreatic cancer, breast cancer, lung cancer, and other cancers and has an impact on the prognosis of cancer; thus, it has become a promising therapeutic target in cancer [108,109,110]. Currently, several PROTACs for histone methylation sites have been studied out (Table 3).

#### 4.3.1. NSD3

Nuclear receptor binding SET domain protein 3 (NSD3) is a well-known lysine methyltransferase (HMTase) and a member of the Nuclear receptor binding SET domain protein family. This family consists of three HMTase enzymes, NSD1, NSD2, and NSD3 [122]. NSD3 drives progression of multiple cancers, mainly through the NSD3–BRD4–Chromodomain-helicase-DNA-binding protein 8(CHD8) axis and the BRD4–NSD3–MYC axis, and is closely associated with poor overall survival and progression-free survival of cancer patients [123,124,125]. In 2022, two NSD3-targeting PROTAC drugs, MS9715 and SYL2158, were designed. Both of these drugs contain NSD3 antagonists linked to E3 ligase VHL ligands. MS9715 is an NSD3 small-molecule degrader based on BI-9321 (a selective antagonist of the NSD3-PWWP1 domain) using PROTAC technology. In acute myeloid leukemia and acute lymphoblastic leukemia blood cancer, MS9715 could more effectively degrade NSD3, specifically inhibit the expression of c-Myc in cancer cells, and affect the growth of cells than BI-9321. Studies have also shown that NSD3-dependent cancers can be treated by MS9715, which is not available in BI-9321. SYL2158 has great selectivity for the degradation of NSD3 and effectively down-regulates the expression of H3K36 methylation and cell proliferation-related genes in lung cancer cells, which is more effective in inhibiting cell growth than NSD3-PWWP antagonist in lung cancer cells. In addition, SYL2158 effectively induced NSD3 degradation in hepatocellular carcinoma xenograft tumor mouse models in liver cancer. In 2022, two NSD3-targeting PROTAC drugs, MS9715 and SYL2158, were designed. Both of these drugs contain NSD3 antagonists linked to E3 ligase VHL ligands. In blood cancer and lung cancer cells, they could effectively degrade NSD3, inhibit the expression of c-Myc in cancer cells, and affect the growth of cells. SYL2158 also reduced NSD3 in xenografted immunodeficient mice [111,112].

#### 4.3.2. PRC2

The polycomb repressive complex 2 (PRC2) is a methyltransferase with a core part consisting of the Enhancer of the zeste homolog 1/2 (EZH1/2), a zinc finger protein polycomb repressive complex 2 subunit (SUZ12) and an embryonic ectoderm development (EED), which catalyzes lysine 27 on histone H3 (H3K27) [126,127]. Numerous studies have shown that PRC2 is dysregulated in human cancers and that its overexpression is often associated with poor prognosis [128]. There have been many inhibitors of PRC2 in clinical use, but clinical data show that the use of these inhibitors will lead to drug resistance. Therefore, PROTACs targeting PRC2 have been developed. The first PCR2-targeting PROTAC was composed of a target protein-binding ligand (EED) and an E3 ubiquitin ligase ligand (VHL) and could degrades PCR2 in cancer cells and inhibit cell proliferation in PRC2-dependent cancer cells. Although this drug did not degrade PCR2 much better than other inhibitors, it indicated a new approach for the treatment of drug-resistant cancers [113]. A subsequent degrader named UNC6852 degraded EED and other PRC2 components in a highly selective manner in HeLa cells and human diffuse large B-cell lymphoma (DB) cells and inhibited PRC2 activity. UNC6852 was much less cytotoxic in DB cells than EED and EZH2 inhibitors, indicating that in some specific cases PROTAC inhibitors have advantages over other inhibitors and providing a direction for further in-depth research in the future [114]. As the core component of PRC2, EZH2 has been proven to have carcinogenic effects in cancer, but current EZH2 inhibitors are not effective in inhibiting the oncogenic activity of EZH2 [129,130]; thus, EZH2-based PROTACs were designed to degrade PRC2 complexes. These PROTACs link a selective EZH2 inhibitor to the CRBN ligand 4-hydroxythalidomide and can degrade the proteasome of the PRC2 subunit in multiple cancer cell types, including EZH2, EED, SUZ12, and Retinoblastoma-binding protein 48 that can completely block EZH2. The carcinogenic activity of EZH2 provides a starting point for the development of carcinogenic active drugs that degrade EZH2 [116]. A VHL-based PROTAC EZH2 degrader was subsequently tested in lymphomas; it was demonstrated to be more efficient than CRBN ligand-based degraders in degrading EZH2, showed advantages over common inhibitors in inhibiting tumor growth in lymphoma xenografts in vivo, and had no obvious toxicity. However, at low concentrations it did not degrade EZH2 well; therefore, this PROTAC still needs to be improved [118]. MS177 and U3i are newly emerging PROTAC drugs targeting EZH2. MS177 could simultaneously degrade the canonical EZH2-PRC2 and non-canonical EZH2-cMyc complexes, targeting the multi-faceted tumorigenic functions of EZH2, with faster onset and inhibition than other inhibitors, resulting in a more marked effect on tumors [115]. U3i could degrade the PRC2 complex in triple-negative breast cancer and induce cancer cell apoptosis, with little damage to normal cells. This PROTAC precisely targets EZH2 in triple-negative breast cancer and is a refined development in the application of PROTAC drugs [117].

#### 4.3.3. PRMT5

Protein arginine methyltransferase 5 (PRMT5) is a type II enzyme of the protein arginine methyltransferase family that methylates the arginine residues of histone and non-histone substrates [131]. It has a very important role in regulating cell proliferation, differentiation, cell cycle progression, DNA damage response, and cell death [132,133,134]. High expression of PRMT5 has been observed in prostate cancer, pancreatic cancer, gastric cancer, etc. Overexpression of PRMT5 is related to the metabolism of cancer cells, can promote the progression and metastasis of cancer, and is related to a poor prognosis in cancer patients [135,136,137,138,139]. Ms4322 is a PROTAC drug based on VHL, which can highly selectively reduce PRMT5 in breast cancer, cervical cancer, lung cancer, leukemia cells, and ER+ breast cancer cells and inhibit the growth of cancer cells. MS4322 has also shown effects in in vitro experiments [119].

#### 4.3.4. WRD5

WDR5 is a highly conserved WD40 repeat protein that is involved in many chromatin-centered processes. Its most prominent role is in the assembly scaffold of the epigenetic “writer” complex, which mediates dimethylation and trimethylation of the histone H3 lysine 4 (H3K4) [140]. WDR5 has been overexpressed in pancreatic cancer, leukemia, colorectal cancer and other cancers [141,142,143,144]. WDR5 plays an important role in the occurrence and development of cancer by affecting the myc pathway, inhibiting DNA damage, inducing the expression of cyclin, and regulating epithelial-mesenchymal transformation [145,146,147]. At present, many studies have shown that wrd5 can affect the metastasis and prognosis of cancer, proving that wrd5 is a potential cancer target [148,149,150].In 2021, Dolle’s team published the first in a series of PROTAC-targeted degradation agents for wrd5. Among them, MS33 is a VHL-based PROTAC drug, which can effectively degrade wrd5 in AML cells and have a good effect on the function of wrd5 [120]. The XUFEN YU team then developed MS67, also a VHL-based PROTAC drug. MS67 is more effective than WDR5 inhibitors in effectively and selectively depleting WDR5, inhibiting transcription of WDR5 regulatory genes, reducing MLL complex components and chromatin binding parts of c-MYC, and inhibiting the proliferation of cancer cells. The team tested MS67 in xenotransplantation mice and found that it was well tolerated and inhibited tumor growth in the mice [121].

### 4.4. Chromatin Remodeling

Chromatin remodeling refers to regulatory changes in the chromatin structure through covalent modification of histones or the action of ATP-dependent remodeling complexes [151]. Chromatin remodelers can be divided into four types: mammalian SWItch/Sucrose NonFermentable (SWI/SNF), mimicSWItch, INO80 Complex ATPase Subunit (INO80), and nucleosome remodeling and deacetylation chromodomain helicase DNA-binding complexes. They have vital roles in replication, cell division, and differentiation and are closely related to the occurrence and development of cancer [152]. Protac drugs targeting chromatin remodeling have been designed and will have great prospects in the treatment of cancer (Table 4).

#### 4.4.1. SMARCA2/SMARCA4

Probable global transcription activator SNF2L2 (SMARCA2) and Transcription activator BRG1 (SMARCA4) are SWI/SNF Adenosine triphosphate enzyme subunits and essential parts of the nuclear body remodeling function of the SWI/SNF complex. Sequencing results have shown that mutations in genes encoding mSWI/SNF occur in more than 20% of cancers [157]. SMARCA4 has an inhibitory effect in solid tumors; however, in AML, it participates in maintaining the oncogenic transcriptional program and promoting proliferation. Therefore, the use of selective inhibitors in inhibiting SMARCA2 activity has become a new therapeutic strategy in SMARCA4-mutant cancers. However, cells lacking SMARCA4 activity are more susceptible to SMARCA2 loss, and SMARCA2/4 inhibitors have not shown antiproliferative effects [158]. ACBI1 is the first SMARCA2/4-targeted protoplasmic PROTAC, formed using a SMARCA bromodomain inhibitor linked to a SMARCA2/4 structure-based VHL E3 ubiquitin ligase ligand. It has been shown to degrade SMARCA2 and SMARCA4 in various cancer cells, deplete SMARCA2 in SMARCA4-mutant AML, and cause decreased cell proliferation and cell death [154]. In 2022, a new SMARCA2- and SMARCA4-targeting PROTAC, AU-15330, was developed. AU-15330 showed preferential toxicity to cancer cells at low doses in enhancer-binding transcription-factor-addicted cancers. When used synergistically with the androgen receptor antagonist enzalutamide, it could induce disease remission in a castration-resistant prostate cancer model. This was the first study to show that modulation of non-coding regulatory elements could be a promising treatment for enhancer-addicted cancers [153].

#### 4.4.2. BRD9

Bromodomain-containing protein 9 (BRD9) is an essential component of the human ATP-dependent chromatin remodeling BRG1/BRM associated factors (BAF) complex. It has been identified as a therapeutic target for subgroups of sarcomas and leukemias [159]. VZ185 is a PROTAC based on a VHL E3 ligase ligand that targets the degradation of BRD9 and its homolog Bromodomain-containing protein 7 (BRD7). BRD9 is a target that cannot be degraded by recruiting E3 ligase VHL; however, the research team has completed the use of E3 ligase VHL to target and degrade BRD4 by continuously exploring the internal spatial structure of the combined PROTAC drug, and the preference for BRD9 is slightly higher than that for BRD7 [155]. The PROTAC for BRD9 FHD-609 has now entered a phase I clinical trial for the treatment of advanced synovial sarcoma.

#### 4.4.3. TRIM24

Tripartite motif 24 (TRIM24), also known as TIF1α, a member of the triplex motif protein family, is a transcriptional regulator that affects the expression and function of chromosome remodeling-related proteins [160,161]. Studies have shown that overexpression of TRIM24 is associated with the progression and poor prognosis of various cancers, including ovarian cancer, liver cancer, and prostate cancer [162,163,164,165]. dTRIM24, a VHL ligand-based PROTAC drug targeting TRIM24, has been shown to induce the rapid and sustained proteasomal degradation of TRIM24, selectively degrading TRIM24 across the entire proteome. In AML cell lines, dTRIM24 affected chromatin, transcription, and proliferation, caused G1/S cell cycle arrest, inhibited cell growth, and promoted apoptosis [156].

#### 4.4.4. ARID1A

ARID1A, a nucleocarp protein, is an SWI/SNF subunit gene that is the most commonly mutated member of the SWI/SNF complex [166]. ARID1A is responsible for several nuclear activities, including transcription, DNA methylation, DNA synthesis, and damage repair [167]. Complexes of ARID1A and BRG1 have been shown to interact and collaborate directly with p53 to transcriptionally regulate multiple downstream effectors [168]. Specific alteration and loss of function of the ARID1A gene are present in about 6% of cancers [169]. A variety of studies have shown that ARID1A can affect the immunotherapy of tumors, the therapeutic effect and prognosis. ARID1A is a potential target for cancer therapy [170,171,172]. PROTAC drugs targeting ARID1A can effectively degrade ARID1A in cells and affect its function, which will play a very good curative effect in the field of cancer therapy. It is a very meaningful target for PROTAC drugs in the field of epigenetic inheritance.

## 5. Summary and Future Prospects

In the past few decades, the development of PROTAC drugs targeting epigenetic modifications has progressed quickly, including improvements in both the structure and the performance of drugs (Figure 2). In terms of targets, the design of PROTACs in the field of epigenetics has mainly concentrated on histone modifications, the targeting of chromatin modification is relatively small, and in the clinical trials of drugs, it would be much less. As a newly developed technology, PROTACs are very promising. Their different degradation mechanism compared with that of inhibitors gives them a unique position in anticancer treatment. In the future, PROTAC drugs targeting epigenetic targets could be used to treat cancer with more precision, and new applications may be discovered.

## Figures and Tables

**Figure 1 molecules-28-01217-f001:**
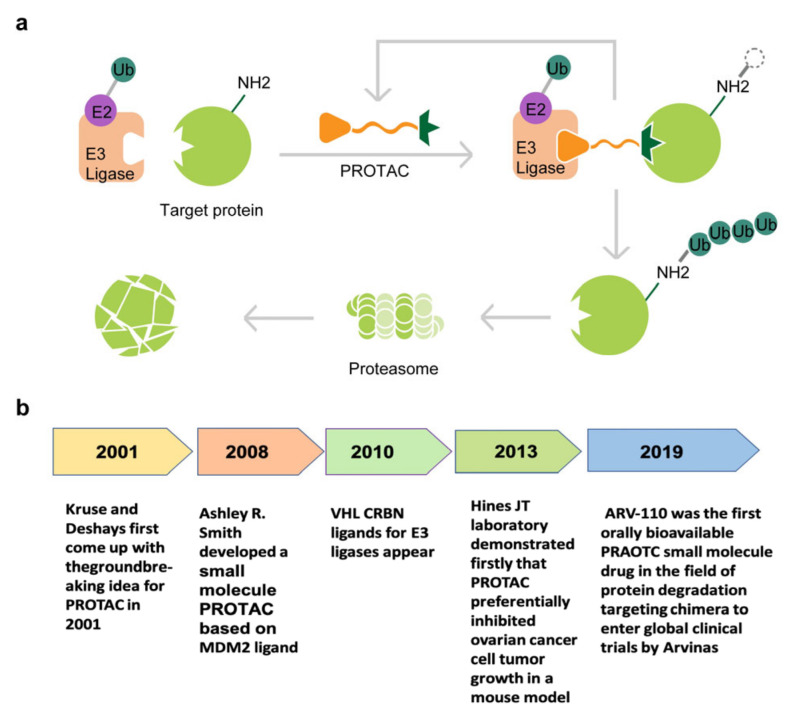
The structure and development process of protac: (**a**) The structure of PROTAC and the mechanism by which it targets protein degradation [8]; (**b**) Major events in the history of PROTAC [9,11,12,13,15].

**Figure 2 molecules-28-01217-f002:**
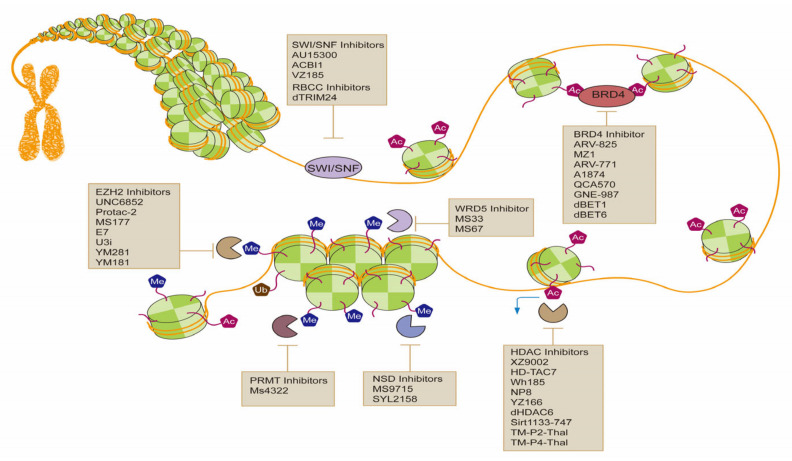
Epigenetic targets in cancer and their PROTA drugs.

**Table 1 molecules-28-01217-t001:** PROTACs for Histone acetylation targets [30,31].

Target	PROTAC	PROTAC Structure	E3 Ligase	Cancer	Ref.
BRD4	ARV-825	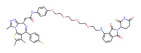	CRBN	Lymphoma Acute myeloid leukemia Neuroblastoma Cholangiocarcinom Thyroid cancer Gastric cancer Ovarian cancer Multiple myeloma Triple-negative breast cancer	[32,33,34,35,36,37,38,39,40,41,42,43,44,45,46,47,48,49]
BRD4	MZ1	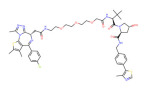	VHL	Triple-negative breast cancer Ovarian cancer Diffuse large B-cell lymphoma Neuroblastoma, Colon cancer HER2-positive breast cancer	[50,51,52,53,54,55]
BRD4	ARV-771	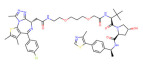	VHL	Prostate cancer Liver cancer Non-small cell lung cancer Esophageal Squamous Cell Carcinoma	[56,57,58,59,60,61,62]
BRD4	dBET1	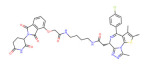	CRBN	Colon cancer, Breast cancer Ovarian cancer Acute Myeloid Leukemia Melanoma	[63,64,65,66]
BRD4	dBET6	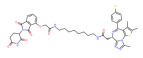	CRBN	Chronic myeloid leukemia Lung cancer Glioblastoma	[67,68,69,70,71]
BRD4	A1874	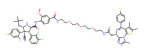	MMD2	Colon cancer Non-small cell lung cancer	[72,73]
BRD4	QCA570	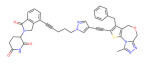	CRBN	Non-small cell lung cancer Acute leukemia	[74,75]
BRD4	GNE-987	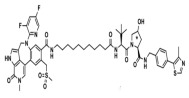	VHL	Acute Myeloid Leukemia	[74,75]
CREBBP	dCBP-1	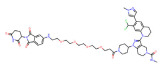	CRBN	Multiple myeloma	[76]

**Table 2 molecules-28-01217-t002:** PROTACs for Histone acetylation targets [30,31].

Target	PROTAC	PROTAC Structure	E3 Ligase	Cancer	Ref.
HDAC6	Wh185	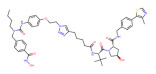	VHL	Multiple myeloma	[82]
HDAC6	YZ166	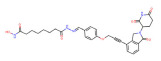	CRBN	Multiple myeloma	[83]
HDAC6	dHDAC6	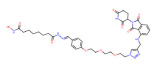	CRBN	Multiple myeloma	[84]
HDAC6	NP8	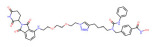	CRBN	Multiple myeloma	[85]
HDAC3	XZ9002	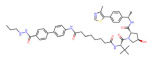	VHL	Breast cancer	[86]
HDAC3	HD-TAC7	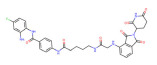	CRBN	Leukaemia	[87]
SIRT2	Sirt1133−747	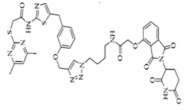	CRBN	Cervical cancer	[88]
SIRT2,	HT7-parkin	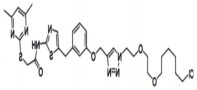	CRBN	Cervical cancer	[89]
SIRT2	TM-P2-Thal	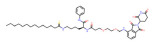	CRBN	Breast cancer	[90]
SIRT2	TM-P4-Thal	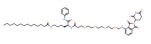	CRBN	Breast cancer	[90]

**Table 3 molecules-28-01217-t003:** PROTACs for Histone methylation targets [30,31].

Target	PROTAC	PROTAC Structure	E3 Ligase	Cancer	Ref.
NSD3	MS9715	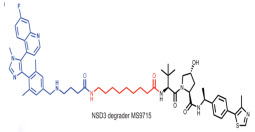	VHL	Blood cancer Lung cancer	[111]
NSD3	SYL2158	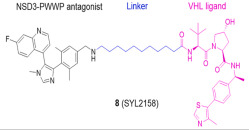	VHL	Blood cancer Lung cancer	[112]
PRC2	PROTAC-2	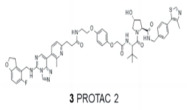	VHL	Diffuse large B-cell lymphoma Rhabdoid cancer	[113]
PRC2	UNC6852	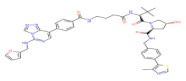	VHL	Cervical cancer Diffuse large B-cell lymphoma	[114]
EZH2	MS177	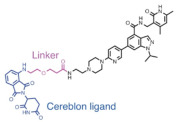	CRBN	MLL-r leukaemias	[115]
EZH2	E7	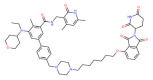	CRBN	Diffuse large B-cell lymphoma Prostate cancer Ovarian cancer	[116]
EZH2	U3i	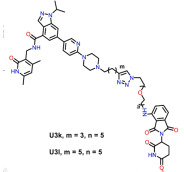	CRBN	Triple-negative breast cancer	[117]
EZH2	YM281	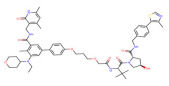	VHL	Lymphoma	[118]
EZH2	YM181	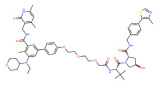	VHL	Lymphoma	[118]
PRMT5	Ms4322	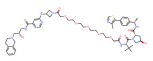	VHL	Breast cancer Cervical cancer Lung cancer Leukemia ER (+) breast cancer	[119]
WRD5	MS33	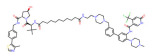	VHL	Acute Myeloid Leukemia	[120]
WRD5	MS67	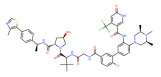	VHL		[121]

**Table 4 molecules-28-01217-t004:** PROTACs for Chromatin remodeling targets [30,31].

Target	PROTAC	PROTAC Structure	E3 Ligase	Cancer	Ref.
SMARCA2/4	AU15300	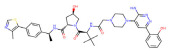	VHL	Prostatic cancer	[153]
SMARCA2/4	ACBI1	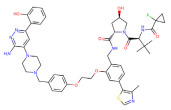	VHL	Non-small cell lung cancer Cutaneous malignant Melanoma Leukemia	[154]
BRD9	VZ185	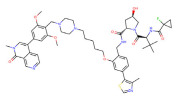	VHL	Cervical cancer	[155]
TRIM24	dTRIM24	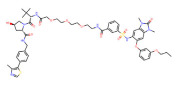	VHL	Acute leukemia	[156]

## Data Availability

PROTAC data is available from PROTACDB (http://cadd.zju.edu.cn/protacdb/about, accessed on 5 October 2022).

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
