# Peer review of "PROTACs in Epigenetic Cancer Therapy: Current Status and Future Opportunities"

_molecules, 2023, doi:10.3390/molecules28031217_

Round 1
Reviewer 1 Report
The manuscript titled PROTACs in Epigenetic Cancer Therapy: Current Status and 2 Future Opportunities" was submitted to the Molecules Journal in the section Medicinal Chemistry.
Liu X.I. and co-authors introduced the readership with the coin epigenetics, epigenetic modifications and the PROTAC strategies to target epigenetic drug targets. In the first paragraph, the authors give an overview of the historical development of PROTACs
- Review prepared by Liu X.I. and co-authors has several points that need to be improved:
- In section 4.1. it should be mentioned that BRD-4 is an epigenetic reader (the focus is to emphasize the terminology "epigenetic reader")
- The structures in Table 1 are hardly observed, please increase the size and clarify them.
- The structure of GNE-987 (Table 2) should be drawn again.
- Section 4.2. replace HADCs with HDACs
- In the sentence “Class III HDAC is a nicotinamide adenine dinucleotide(NAD)-dependent enzyme” please replace with NAD+-dependent histone deacetylases (use NAD+ as an oxidized form of NAD).
- Line 221 - Please add the word representative at the end of the sentence - HDAC11 is the only class IV HDAC “representative”.
- Please, increase the size of the structures in Table 3, Table 4, Table 5. and use the same style for chemical structure representation.
- The summary should be discussed with more details regarding the classes of epigenetic targets that could be targeted with PROTACs.
Author Response
Please see the attachmen

Reviewer 2 Report
The review gives a comprehensive summary across the currently published PROTACs covering epigenetic targets. Individual degraders are highlighted and their discovery and design discussed in detail only for the Histone acetylation (eg. Bromodomain) family whereas other target classes are covered in less detail. An exploration in more detail on these other classes could improve the review further and differentiate it from providing mostly a database resource (like ProtacDB w/ a header description for each target class).
All tables contains spelling errors and arbitrary capitalisation in the header "TRAGET", "RAF", "CANCER" (not an abbreviation; should not be all-caps).
All tables should be checked for consistency in how structures are displayed (eg. p6 Table2 GNE987 PROTAC structure appears distorted, several structures contain additional text eg. p10 Table 4 Protac-2, MS177, U3i and appear to have been copied out of papers).
Reviewer 3 Report
The paper is about applying the PROTAC strategy to Cancer.
The authors wrote a brief review regarding the PROTAC strategy, which is too short in its current form. There are some major shortcomings that need to address.
1. Authors point out the PROTAC strategy as an epigenetic way of targeting cancer, which is true for most PROATC compounds as they target intracellular proteins; then why did the author choose only a few PROTACs?
There are more than 400 Protacs scaffolds available, please see the following papers.
(2022). PROTAC targeted protein degraders: the past is prologue. Nature Reviews Drug Discovery, 21(3), 181-200.
(2021). Advancing targeted protein degradation for cancer therapy. Nature Reviews Cancer, 21(10), 638-654.
(2022). PROTACs: past, present and future. Chemical Society Reviews.
(2020). PROteolysis TArgeting Chimeras (PROTACs) as emerging anticancer therapeutics. Oncogene, 39(26), 4909-4924.
(2019). Development of targeted protein degradation therapeutics. Nature chemical biology, 15(10), 937-944.
(2022). Strategies to Reduce the On‐Target Platelet Toxicity of Bcl‐xL Inhibitors: PROTACs, SNIPERs and Prodrug‐Based Approaches. ChemBioChem, e202100689.
and many more...
2. There are over 80 reviews published in the last 4 years on "PROTACs in Cancer" How the current review is different than other reviews published in this field?
Minor Comments
1. Typographical error, the template used for the current paper is of "MDPI CHEMISTRY" journal, while the First page is used from "MDPI MOLECULES"
Unfortunately, the current paper doesn't seem to appeal in the interest of readers in the field of PROTAC targetting. The author needs to specify the PROTACs either based on their target approach.
[such as kinases ((2021). Targeting protein kinases degradation by PROTACs. Frontiers in Chemistry, 9.), bromodomain ((2022). BET bromodomain inhibitors. Current Opinion in Chemical Biology, 68, 102148.), estrogen receptor ((2022). Estrogen Receptor-α Targeting: PROTACs, SNIPERs, Peptide-PROTACs, Antibody Conjugated PROTACs and SNIPERs. Pharmaceutics, 14(11), 2523.)), and other common targets of PROTACs.
Another way around this is a hybrid approach, such as Antibody conjugated PROTACs ((2020). Antibody–PROTAC conjugates enable HER2-dependent targeted protein degradation of BRD4. ACS chemical biology, 15(6), 1306-1312.), Light activating PROTACs ((2022). Light-Activating PROTACs in Cancer: Chemical Design, Challenges, and Applications. Applied Sciences, 12(19), 9674.)
Round 2
Reviewer 3 Report
The authors revised the article.